# Rank-Preserving Calibration of LLMs Under Model and Distribution Shifts

## Abstract

A central barrier to deploying Large Language Models (LLMs) in safety-critical applications is hallucination, where models generate non-factual content with high confidence. Detecting hallucinations requires well-calibrated confidence estimates, yet calibration is brittle under domain and model shifts. The former renders confidence estimates unreliable in a new environment, while the latter arises because different LLMs exhibit distinct confidence scales, so calibration learned for one model often fails to transfer when another is used at deployment for efficiency or privacy. Addressing this vulnerability is critical for robust model generalisation, as failure to calibrate reliably confidence values across domains and models undermines trust in LLMs at deployment. Existing prompting-based approaches are label-free and flexible, they perform poorly when domain knowledge of a model is limited. In contrast, explicit calibration for a specialized domain achieves strong in-domain results but fails to generalize to a novel domain. This work discovers although absolute confidence values often fail to transfer across shifts, their *relative rankings*, which only rely on the relative reliability among samples within a dataset, can prevail in robustness across shifts. Based on this key insight, we propose a two-stage framework **R**ank-preserving **A**daptive **P**seudo-**Cal**ibration (RAPCal). In source calibration, an Expectation-Maximisation stage converts one-hot correctness labels into soft supervision by bin-wise accuracy estimations for a fine-grained calibration. In target calibration, a stage for preserved ranking of confidence scores is introduced to construct pseudo soft labels, enabling unsupervised cross-domain calibration adaptation without ground-truth labels in test domains. Experiments show that RAPCal reduces ECE by 6.15% without sacrificing task performance, advancing the reliability of LLMs in label-scarce settings. Code is given in the supplementary materials.

## 1 Introduction

Large language models (LLMs) are effective in performing in various question answering (QA) and reasoning tasks (Brown et al., 2020; Chowdhery et al., 2022; OpenAI, 2023). Yet, *hallucination* remains a persistent and critical challenge. Hallucinations occur when LLMs generate fluent but non-factual content, often with high confidence, undermining their trustworthiness in real-world applications (Ji et al., 2023; Huang et al., 2023). To mitigate this issue, recent works have focused on calibration, i.e., aligning predicted model inference confidence with the true likelihood of inference correctness (Guo et al., 2017; Kumar et al., 2019). Existing methods fall into two categories: (1) Test-time methods, such as temperature scaling (Guo et al., 2017), self-consistency (Wang et al., 2023), and self-reflection (Shinn et al., 2023), adjust predictions at inference but remain constrained by model priors and struggle with novel knowledge. (2) Training-based methods instead leverage one-hot correctness labels as auxiliary supervision, which can improve calibration more effectively (Kuleshov et al., 2018). Training-based methods perform well when both the labelled training data (source) and unlabelled test data (target) are in the same distribution. In this case, correctness labels in a domain provide direct supervision for calibration across training and test data, and the trained model generalises well, as expected, to test data in the same domain (Kuleshov et al., 2018).

In practice, in deployment, test samples often come from a shifted independent target domain without correctness labels. As a result, a model calibrated on a labelled source domain no longer transfers reliably to an unlabelled target domain of a different distribution, resulting in adopting in-domain

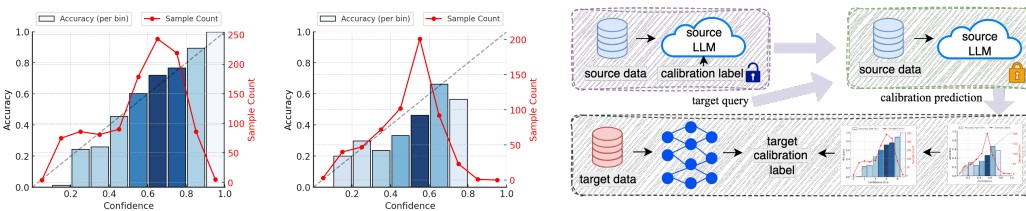

(a) Source calibration        (b) Target calibration        (c) Illustration of our new setting

Figure 1: **Rational.** (a) On the source-domain TriviaQA dataset, supervised calibration yields confidence scores that closely track accuracy, with an ECE of 4.96%. (b) Under a distribution shift setting, calibration degrades: on the target-domain SciQA dataset, the ECE deteriorates to 16.63%, confidence values collapse toward the mid-range, and sample counts cluster in central bins, making the scores unreliable. (c) Given both model and distribution shifts across different domains, RAPCal first learns supervised *direct* calibration with labelled source data, followed by ranking-preserved unsupervised *relative* calibration adaptation on unlabelled sparse target data, whilst the source model is frozen with only its 'black-box' outputs available as calibration adaptation guidance.

calibration adaptation invalid (Ovadia et al., 2019; Guo et al., 2017). As shown in reliability diagrams of Fig. 1, in-domain calibration yields confidence scores that align with accuracy reasonably well, whereas in a target domain from a different distribution, confidence collapses toward the mid-range (around 0.5). The red curves further show that predictions concentrate in the middle bins, indicating that the model avoids confident estimates and fails to provide effective calibration (Hendrycks and Gimpel, 2017). Ideally, cross-domain calibration would require adapting a source-trained model to a target distribution. In reality, a source model is typically inaccessible as a 'black box': either locked as a frozen checkpoint, exposed only through APIs, or restricted by privacy, security, or commercial constraints (Tramèr et al., 2016; Liang et al., 2020b). This prevents parameter updates, leaving only black-box outputs such as predictions and confidence scores for supervision. A naive alternative is to transfer source-trained calibration heads, but: (i) parameters fitted to the source distribution often fail to generalise, and (ii) target users may rely on a different backbone, e.g. due to restricted budgets, making strict parameter transfer impractical (Guo and Rush, 2022; Zhang et al., 2021). Solving this problem is both significant to LLMs when deployed in practice, and challenging as contemporary LLMs confidence calibration methods fail to address it effectively.

This paper proposes a setting where a calibration head is first learned in a labelled source domain, but once trained the source model checkpoint is frozen and inaccessible. During adaptation to the unlabelled target domain, only black-box source outputs such as predictions and confidence scores are available (See Fig. 1(c)). In this setting, directly transferring source-domain calibration predictions to the target domain is infeasible, since distribution and model shifts distort confidence values. However, *rankings comparing the relative reliability of samples within each domain* is stable even under such cross-domain shifts. Rather than transferring weights, our goal is therefore to transfer calibration strategies, enabling unsupervised calibration refinement even under distribution shift and heterogeneous model backbones.

More specifically, we introduce RAPCal, a two-stage approach to cross-domain inference confidence calibration adaptation, a two-stage framework that achieves robust calibration under both distribution and model shifts. A lightweight calibration head, implemented as a small neural module that maps model logits into calibrated probabilities, is first learned on a labelled source domain and then adapted to an unlabelled target domain. (i) In the source pre-calibration stage, we introduce an EM-based strategy that converts one-hot correctness labels into soft supervision. The E-step estimates bin-wise accuracy from the confidence–accuracy curve, while the M-step updates the calibration head with these refined labels. This iterative refinement, starting from one-hot labels, equips the model with fine-grained relative ranking values for in-domain calibration. (ii) In the target unsupervised calibration stage, where no correctness labels are available, we exploit the preserved ranking of confidence scores from the source-pretrained calibration head. By transforming ranks into pseudo soft labels and combining them with raw confidence outputs, the model learns to map source calibration to a target domain by preserving the relative rankings of source confidence subject to target domain shift without ground-truth labels. Our main contributions are threefold:

1. We formalise a novel approach to solving a cross-domain and cross-model inference confidence calibration problem where a target domain has no ground-truth labels and a pre-trained source do-

main model is only available as a 'black-box'. This reflects more truthfully the challenge in practical deployments where LLMs are frequently accessed only by APIs and it becomes nontrivial to retain robust confidence calibration under distribution shifts without direct model access.

2. We propose RAPCal, a two-stage framework that combines EM-based fine-grained calibration in a source domain with rank-preserving pseudo-label refinement in a target domain. Our key insight is whilst absolute confidence scores fail to transfer across models and domains, their relative ranking remains stable, allowing RAPCal to adapt calibration fully unsupervised given a 'black-box'.

3. Extensive experiments on multiple QA benchmarks show that RAPCal consistently improves calibration under both model and domain shifts while preserving model cross-domain inference performance. It is demonstrated that reliable confidence estimation is achievable even in label-free and model-shifted deployment scenarios, addressing a critical barrier for the safe and more trustworthy use of LLMs.

## 2 RELATED WORKS

**Calibration.** Calibration aims to align model confidence with the true likelihood of correctness, thereby mitigating overconfidence and hallucination in large language models (LLMs). Classical approaches include post-hoc methods such as Platt scaling (Platt, 1999; Guo et al., 2017) and isotonic regression (Zadrozny and Elkan, 2002), which adjust confidence at inference time. More recent efforts leverage LLM-specific properties, including self-consistency (Wang et al., 2022), majority voting (Kadavath et al., 2022), or self-reflection (Shinn et al., 2023), to improve reliability without training. However, such test-time methods remain constrained by model priors and often fail to handle a specialised domain. Training-based calibration methods instead introduce auxiliary supervision using correctness labels. For instance, token-level or sequence-level correctness signals are used to train confidence heads (Moon et al., 2020; Jiang et al., 2021; Zhang et al., 2025). These methods achieve more effective calibration but rely on the availability of ground-truth correctness labels in a deployment target domain. In practice, this assumption is restrictive: correctness labels are expensive to obtain, especially for open-ended QA, and are typically absent in deployment. Moreover, most existing methods assume 'white-box' access to the model for fine-tuning, unrealistic when LLMs are accessed through APIs or released as frozen checkpoints 'black boxes'.

**Domain Adaptation.** Domain adaptation seeks to transfer model performance from a labelled source domain to an unlabelled target domain under distribution shift (Ganin et al., 2016; Long et al., 2018). Classical approaches include discrepancy minimisation (Long et al., 2015; Sun and Saenko, 2016), adversarial alignment (Ganin et al., 2016; Long et al., 2018), and pseudo-label self-training (Lee, 2013; Liang et al., 2020a). More recent studies in black-box domain adaptation (BBDA) (Liang et al., 2020a; Kundu et al., 2020) restrict access to the source model's parameters, relying only on its outputs for adaptation. This setting is particularly relevant to modern LLMs, where source models are often only accessible in a black-box form. Despite extensive work in domain adaptation, calibration under domain shift has received little attention. Existing calibration methods are predominantly in-domain and supervised, while existing domain adaptation methods focus on classification tasks with closed label spaces (Guo et al., 2017; Wang et al., 2020; Chen et al., 2021). In contrast, LLMs pose unique challenges: outputs are open-ended, correctness labels are unavailable in the target domain, and calibration must be achieved without modifying the source model. We bridge these two lines of research by introducing *confidence calibration across model and distribution shifts*. Our approach leverages correctness labels in the source domain to equip the model with fine-grained calibration ability, and then transfers this ability to the unlabelled target domain by exploiting rank statistics of model outputs. This design directly addresses the limitations of prior works: it avoids reliance on target-domain labels, respects the black-box constraint of modern LLMs, and remains effective under distribution shift.

## 3 RANK-PRESERVING ADAPTIVE PSEUDO-CALIBRATION

This section introduces the proposed Rank-Preserving Adaptive Pseudo-Calibration (RAPCal) framework. We first outline the problem formulation (§3.1), then describe the source pre-calibration stage with EM refinement (§3.2.1), and the target unsupervised calibration procedure (§3.2.2).

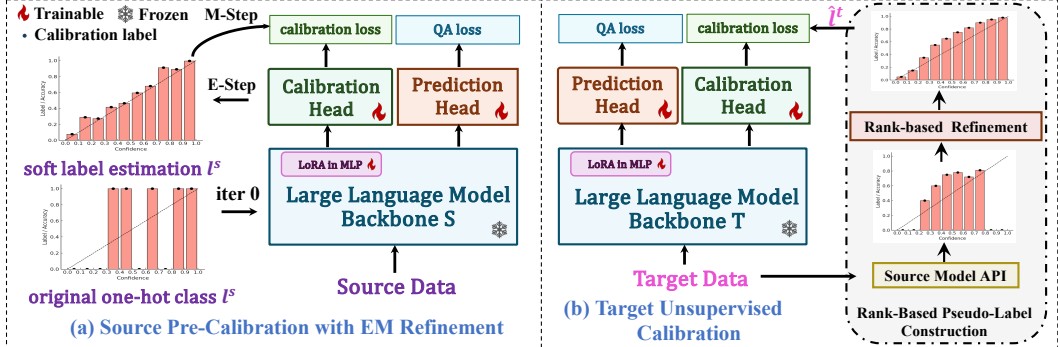

Figure 2: RAPCal overview. (a) **Source Pre-Calibration with EM Refinement**: Source data train LoRA modules (inserted into MLP layers of the frozen backbone), a prediction head, and a calibration head. The EM procedure iteratively refines calibration supervision: the E-step estimates bin-wise accuracies from correctness labels, in the M-step the calibration head is updated with these soft labels using calibration loss $\mathcal{L}_c$, while QA ability is preserved via QA loss $\mathcal{L}_{\text{qa}}$. (b) **Target Unsupervised Calibration**: On target data, empirical ranks from the source-trained calibration head are transformed with a U-shaped refinement and interpolated with raw scores to construct robust pseudo calibration labels $\tilde{l}^t$ through rank-based pseudo label construction. Then the calibration head and prediction head are optimised on target data using $\mathcal{L}_c$ and $\mathcal{L}_{\text{qa}}$, respectively. The backbone remains frozen (blue), while lightweight modules (green, purple, orange) trainable.

### 3.1 PRELIMINARY

We study LLM calibration under a **model and distribution shifts**. A labelled source domain is denoted as $\mathcal{D}_s = \{X_i^s, \hat{Y}_i^s, l_i^s\}$, where $X_i^s \in \mathcal{X}$ is the $i$th input query, the LLM generates answers $\hat{Y}_i^s = \{\hat{y}_{i1}^s, \ldots, \hat{y}_{im_i}^s\}$ with $m_i$ tokens, and $l_i^s \in \{0, 1\}$ provides correctness labels. During training on $\mathcal{D}_s$, we assume white-box access so that both the backbone and an attached calibration head can be optimised, equipping the model with in-domain confidence calibration. An unlabelled target domain is denoted as $\mathcal{D}_t = \{X_i^t, \hat{Y}_i^t\}$, where $X_i^t$ is the $i$th queries and the LLM generates answers $\hat{Y}_j^t = \{\hat{y}_{i1}^t, \ldots, \hat{y}_{im_i}^t\}$. The target domain does not share the source domain's distribution (domain shift). In cross-domain calibration, correctness labels are unavailable in the target domain and source model parameters are also not accessible: only its predictions and confidence scores can be used in a black-box manner. To adapt, we initialise a new calibration head for target domain $\mathcal{D}_t$, which learns solely from outputs of the source model. Note that this model-agnostic design mirrors realistic deployment, where source models are distributed as frozen checkpoints or APIs, Moreover, we allow calibration adaptation when the target domain employs a different backbone (model shift). We aim to transfer calibration ability learned in $\mathcal{D}_s$ to $\mathcal{D}_t$, ensuring reliable confidence estimation while retaining source domain performance.

### 3.2 METHODOLOGY

Our framework consists of two stages: (1) *Source pre-calibration*, which equips the model with improved calibration ability in $\mathcal{D}_s$, and (2) *Target unsupervised calibration*, which transfers calibration from the source model and domain to the target using only the model's outputs.

#### 3.2.1 SOURCE PRE-CALIBRATION WITH EM REFINEMENT

We attach a lightweight calibration head to the backbone of the source model. The calibration head is a two-layer MLP that takes the averaged hidden representation $\bar{z}_i$ of the predicted answer $\hat{y}_i$ as input and outputs a confidence score $c_i \in [0, 1]$:

$$c_i = \sigma(W^\top \bar{z}_i + b), \quad \bar{z}_i = \frac{1}{m_i} \sum_{k=1}^{m_i} z_{ik}, \qquad (3.1)$$

where $z_{ik}$ are token-level hidden states of $\hat{y}_i$ and $m_i$ is its length.

Existing calibration uses one-hot correctness labels $l_i \in \{0, 1\}$ as supervision (Wang et al., 2020; Zhang et al., 2025). However, these binary labels provide only coarse signals, limiting the ability to learn well-calibrated probabilities. To overcome this problem, we introduce an **EM-based refinement** mechanism that iteratively maps binary hard labels to soft estimates of probability values.

**Initialisation (iteration 0).** The calibration head is first trained on one-hot correctness labels $l_i^s$ from $\mathcal{D}_s$, producing initial scores $c_i^s$.

**E-step.** Given the initial confidence scores $c_i^s$, predictions are partitioned into bins, and the empirical accuracy of each bin is estimated. For a sample $i$ falling into bin $b(i)$, we assign a refined soft label

$$\tilde{l}_i^s = \text{Acc}(b(i)), \tag{3.2}$$

where $\text{Acc}(b(i))$ is the average correctness of all samples in bin $b(i)$. Thus, $\tilde{l}_i^s \in [0,1]$ represents the estimated probability that predictions in the same confidence range are correct. These bin-level probabilities replace the original binary labels, providing richer and more stable supervision.

**M-step.** The source model is updated using $\tilde{l}_i^s$ as targets. The overall loss combines calibration and QA preservation:

$$\mathcal{L} = \mathcal{L}_C(c_i^s, \tilde{l}_i^s) + \gamma\, \mathcal{L}_{qa}(X^s, \hat{Y}^s) = \frac{1}{|\mathcal{D}_s|} \sum_i (c_i^s - \tilde{l}_i^s)^2 - \gamma \frac{1}{|\mathcal{D}_s|} \sum_i \log p_\theta(\hat{y}_i^s | x_i^s), \tag{3.3}$$

where $\mathcal{L}_C$ ensures calibration accuracy and $\mathcal{L}_{qa}$ prevents degradation of QA ability. By alternating E- and M-steps, the model gradually refines coarse $\{0,1\}$ supervision into bin-level probabilities, improving calibration granularity and stability.

### 3.2.2 Target Unsupervised Calibration

**Rank-Based Pseudo-Label Construction.** Although the EM refinement yields reliable calibration in $\mathcal{D}_s$, directly applying the source-trained head to predict $\mathcal{D}_t$ often causes mid-range collapse, where most confidence scores cluster around $0.5$ as Fig. 1 shows. Absolute values often degrade under distribution shift, but the *relative ranking* of predictions usually remains stable and robustly informative in relative terms: higher ranks tend to correspond to correct predictions, while lower ranks are more likely incorrect. To exploit this, we construct pseudo labels $\tilde{l}_i^t$ that preserve ranking information while countering mid-range collapse. For each target prediction, the empirical rank is as

$$r_i^t = \frac{\text{rank}(c_i^t) + 0.5}{N_t}, \tag{3.4}$$

where $N_t = |\mathcal{D}_t|$ and $\text{rank}(\cdot)$ returns the ascending index of $c_i^t$. This yields $r_i^t \in (0,1)$ for every sample. Ranks are then pushed toward the extremes by a U-shaped mapping:

$$p_{i,\text{shaped}}^t = \begin{cases} 0.5 \cdot (2r_i^t)^\alpha, & r_i^t < 0.5, \\ 1 - 0.5 \cdot (2(1 - r_i^t))^\alpha, & r_i^t \geq 0.5, \end{cases} \tag{3.5}$$

where $\alpha > 1$ controls the stretching strength. By construction, $p_{i,\text{shaped}}^t \in [0,1]$ and increases the separation between high- and low-ranked predictions. Finally, to retain information from the raw confidence $c_i^t$, we interpolate:

$$\tilde{l}_i^t = \lambda\, c_i^t + (1 - \lambda)\, p_{i,\text{shaped}}^t, \tag{3.6}$$

where $\lambda \in [0,1]$ balances reliance on raw calibration and rank-based refinement. Thus, $\tilde{l}_i^t$ also lies within $[0,1]$, serving as continuous pseudo supervision.

This design ensures pseudo labels are bounded and interpretable: they preserve ordering from the source-informed calibration head while amplifying reliable extremes. In ablation, we further verify that replacing rank-based refinement with permuted ranks degrades performance, validating the benefit of informative ranking, rather than merely scattering probabilities.

**Target Calibration.** In $\mathcal{D}_t$, the source backbone is frozen and only black-box outputs are accessible. A new calibration head, with the same lightweight MLP architecture as in the source stage, is randomly initialised and trained on top of the frozen backbone representations. For each sample, the head produces a logit $\ell_i^t$, which is further adjusted by a constant bias $\tau \in \mathbb{R}$ and a temperature $T > 0$:

$$\hat{c}_i^t = \sigma\left(\frac{\ell_i^t + \tau}{T}\right). \tag{3.7}$$

Here, $\tau$ shifts the logits, while $T$ adjusts their sharpness for confidence control. The training objective combines pseudo-label supervision with QA preservation:

$$\mathcal{L}_t = \mathcal{L}_C(\hat{c}^t, \tilde{l}^t) + \gamma\,\mathcal{L}_{qa}(X^t, \hat{Y}^t) = \frac{1}{|\mathcal{D}_t|}\sum_{i=1}^{|\mathcal{D}_t|}(\hat{c}_i^t - \tilde{l}_i^t)^2 - \gamma\frac{1}{|\mathcal{D}_t|}\sum_{i=1}^{|\mathcal{D}_t|}\log p_\theta(\hat{y}_i^t|x_i^t), \qquad (3.8)$$

where $\tilde{l}_i^t$ are the rank-based pseudo labels and $\mathcal{L}_{qa}$ ensures QA ability is retained. This design allows calibration to adapt in $\mathcal{D}_t$ without requiring target labels or access to source parameters, while the frozen backbone guarantees black-box compatibility.

## 4 THEORETICAL ANALYSIS

In this section, we analyse the theoretical connection between the rank-pushing function and reliability of the pseudo labels after the rank-pushing function.

**Theorem 4.1** (Cross-domain rank validity under domain shift)**.** *Let the target correctness be* $\eta_t(x) = \Pr_{\mathcal{D}_t}(C = 1 \mid x)$ *and the head output be* $c_t(x) \in [0,1]$*. Assume there exists a one-dimensional* latent order $\ell(x)$ *with strictly increasing maps* $\psi_t^*, g_t$ *such that* $\eta_t(x) = \psi_t^*(\ell(x))$ *and* $c_t(x) = g_t(\ell(x)) + \varepsilon_t(x)$*, where the shift perturbations* $\varepsilon_t(x)$ *are independent, mean-zero, sub-Gaussian with variance proxy* $\sigma^2$*. If* $g_t$ *is* $m$*-strongly monotone (i.e.,* $g_t(u_1) - g_t(u_2) \geq m(u_1 - u_2)$ *for* $u_1 > u_2$*, equivalently* $m = \inf_u g_t'(u) > 0$ *when differentiable), then for any pair with latent gap* $\Delta_{ij} := \ell(x_i) - \ell(x_j) > 0$*,*

$$\Pr\big(c_t(x_i) \leq c_t(x_j)\big) \;\leq\; \exp\Big(-\,\frac{m^2\,\Delta_{ij}^2}{4\,\sigma^2}\Big).$$

*Hence, pairwise rank flips are exponentially rare and the ranking by* $\{c_t(x)\}$ *is reliable with respect to the correctness order.*

*Proof.* Fix $x_i, x_j$ with $\Delta_{ij} > 0$. Then

$$c_t(x_i) - c_t(x_j) = \big[g_t(\ell(x_i)) - g_t(\ell(x_j))\big] + \big(\varepsilon_t(x_i) - \varepsilon_t(x_j)\big).$$

By $m$-strong monotonicity, $g_t(\ell(x_i)) - g_t(\ell(x_j)) \geq m\,\Delta_{ij}$. Let $\xi_{ij} := \varepsilon_t(x_i) - \varepsilon_t(x_j)$. Independence and sub-Gaussianity imply $\xi_{ij}$ is mean-zero sub-Gaussian with variance proxy $\leq 2\sigma^2$ since $\mathbb{E}[e^{\lambda\xi_{ij}}] \leq e^{\lambda^2(2\sigma^2)/2}$. A flip requires $\xi_{ij} \leq -m\Delta_{ij}$, so by the standard sub-Gaussian tail bound,

$$\Pr\big(c_t(x_i) \leq c_t(x_j)\big) \leq \exp\Big(-\,\frac{(m\Delta_{ij})^2}{2 \cdot 2\sigma^2}\Big) = \exp\Big(-\,\frac{m^2\Delta_{ij}^2}{4\sigma^2}\Big).$$

Since $\psi_t^*$ is strictly increasing, ordering by $\ell$ matches ordering by $\eta_t$, so the bound is with respect to the correctness order. $\square$

*Interpretation.* The bound $\exp\big(-m^2\Delta_{ij}^2/(4\sigma^2)\big)$ depends on: (i) the latent gap $\Delta_{ij}$ (how separated the pair is), (ii) the link strength $m$, (iii) the shift scale $\sigma$ (noise level). Larger $\Delta_{ij}$ or $m$, or smaller $\sigma$, make rank flips exponentially less likely. The rank-pushing function can increase the Lipschitz constant $m$ to reduce the probability of unreliable confidence scores after a distribution shift.

## 5 EXPERIMENTS

In this section, we conduct cross-domain QA calibration experiments to demonstrate the effectiveness of our RAPCal. Specifically, we evaluate RAPCal in open-ended QA generation calibration.

### 5.1 SETUP

**Experimental Settings.** Unlike the multiple-choice QA setting, where candidate answers are provided, the open-ended QA generation setting requires the model to directly produce answers (in short phrases) that match the ground truth, using greedy decoding. We adopt two representative datasets, *TriviaQA* (Joshi et al., 2017) and *SciQA* (Welbl et al., 2017), as our primary benchmarks. In our cross-domain QA calibration setup, we treat one dataset as the labelled source domain and the other as the unlabelled target domain. The model is trained on the source domain to simultaneously preserve its QA capability and learn in-domain calibration using correctness labels. It is then adapted to the unlabelled target domain, where it must maintain QA performance while achieving

| Methods | TriviaQA | | | SciQA | | |
|---|---|---|---|---|---|---|
| | ECE ↓ | Brier ↓ | AUROC ↑ | ECE ↓ | Brier ↓ | AUROC ↑ |
| **In-domain Setting (Supervised Learning)** | | | | | | |
| Seq. Likelihood (Kamath et al., 2020) | 12.56 | 21.88 | 73.27 | 16.52 | 26.74 | 61.41 |
| Platt Scaling (Platt, 1999) | 13.95 | 23.47 | 73.27 | 10.12 | 24.84 | 61.41 |
| P(True) (Kadavath et al., 2022) | 9.82 | 23.12 | 75.52 | 7.46 | 20.89 | 69.61 |
| Verbal (Kadavath et al., 2022) | 34.31 | 37.02 | 50.09 | 56.05 | 37.94 | 53.07 |
| Apricot (Zhao et al., 2021) | 11.88 | **15.62** | **81.52** | **6.36** | **16.97** | **80.90** |
| ActCab (Si et al., 2022) | 5.18 | 23.75 | 56.21 | 9.34 | 24.51 | 53.00 |
| **Cross-domain Setting** | | | | | | |
| *(a) Direct Generation* | SciQA → TriviaQA | | | TriviaQA → SciQA | | |
| Apricot (Zhao et al., 2021) | 12.78 | 23.90 | 66.28 | 17.43 | 23.80 | 65.37 |
| ActCab (Si et al., 2022) | 15.75 | 25.78 | 54.89 | 18.83 | 28.77 | 55.93 |
| *(b) Adaptation* | SciQA → TriviaQA | | | TriviaQA → SciQA | | |
| RAPCal (Ours) | **9.60** | **20.85** | **71.15** | **5.97** | **22.92** | **66.28** |

Table 1: Performance comparison for Llama2-7B on the open-ended generation calibration task. We divide methods into **In-domain supervised learning** and **Cross-domain setting**, where the latter includes *(a) Direct generation without adaptation* and *(b) Adaptation*.

| Methods | TriviaQA | | | SciQA | | |
|---|---|---|---|---|---|---|
| | ECE ↓ | Brier ↓ | AUROC ↑ | ECE ↓ | Brier ↓ | AUROC ↑ |
| **In-domain Setting (Supervised Learning)** | | | | | | |
| Seq. Likelihood (Kamath et al., 2020) | 14.57 | 20.62 | 81.45 | 14.47 | 23.81 | 73.89 |
| Platt Scaling (Platt, 1999) | 10.75 | 22.43 | 81.45 | 10.83 | 24.57 | 73.69 |
| P(True) (Kadavath et al., 2022) | 12.65 | 17.85 | **81.69** | 31.82 | 30.09 | **79.66** |
| Verbal (Kadavath et al., 2022) | 27.34 | 28.96 | 63.01 | 36.52 | 36.55 | 56.44 |
| Apricot (Zhao et al., 2021) | 9.69 | 17.85 | 78.36 | **7.09** | **18.80** | 76.45 |
| ActCab (Si et al., 2022) | **8.47** | 22.46 | 64.27 | 8.72 | 22.20 | 70.24 |
| **Cross-domain Setting** | | | | | | |
| *(a) Direct Generation* | SciQA → TriviaQA | | | TriviaQA → SciQA | | |
| Apricot (Zhao et al., 2021) | 13.16 | 26.54 | 61.72 | 18.54 | 28.70 | 52.02 |
| ActCab (Si et al., 2022) | 14.85 | 25.19 | 58.32 | 17.74 | 29.06 | 53.89 |
| *(b) Adaptation* | SciQA → TriviaQA | | | TriviaQA → SciQA | | |
| RAPCal (Ours) | **10.13** | **21.14** | **65.53** | **5.61** | **24.68** | **57.13** |

Table 2: Performance of Phi3-3.8B on the open-ended generation calibration task. Methods are grouped into **In-domain** and **Cross-domain** setting, with the latter split into *(a) Direct generation* and *(b) Adaptation (ours)*. Lower ECE/Brier and higher AUROC indicate better calibration.

cross-domain calibration. To better assess the effectiveness of our method, we compare it against two baselines: Seq. Likelihood, Platt Scaling, P(True), Verbal, Apricot (Zhao et al., 2021), ActCab (Si et al., 2022) , and (ii) domain generalisation approaches that train SOTA methods Apricot (Zhao et al., 2021) and ActCab (Si et al., 2022) solely on the source domain and directly generalise to the target domain. Notably, prior cross-domain calibration methods have been developed for closed-set classification tasks, where the label spaces of the source and target domains are aligned. Such approaches are not applicable to open-ended QA calibration, and thus are not included in our comparisons. Finally, to evaluate the robustness of our method, we conduct experiments with two different backbone models: *Phi-3 3.8B* (Abdin et al., 2024) and *Llama-2 7B* (Touvron et al., 2023).

**Metrics.** We evaluate calibration performance using standard metrics, including expected calibration error (ECE) [50], Brier score [51], and AUROC. ECE quantifies the average mismatch between predicted confidence and actual accuracy across discrete confidence bins, with the number of bins fixed at 10 in all experiments. The Brier score captures the mean squared difference between predicted confidence scores and ground-truth correctness. AUROC reflects how well the confidence scores indicate the factuality of the predicted answers.

(a) Supervised calibration on TriviaQA with our method (in-domain).

(b) Cross-domain generalisation from SciQA → TriviaQA with Apricot.

(c) Cross-domain generalisation from SciQA → TriviaQA with RAPCal.

(d) Cross-domain adaptation from SciQA → TriviaQA with RAPCal.

Figure 3: Reliability diagrams on TriviaQA under different calibration settings. (a) In-domain supervised calibration with RAPCal achieves effective alignment. (b–c) Cross-domain generalisation from SciQA → TriviaQA with Apricot and RAPCal both suffer mid-range collapse. (d) Cross-domain adaptation with RAPCal restores reliable calibration and improves confidence alignment.

| source model: Phi-3.8B | | | target model: Phi-3.8B | | |
| --- | --- | --- | --- | --- | --- |
| SciQA → TriviaQA | | | TriviaQA → SciQA | | |
| ECE ↓ | Brier ↓ | AUROC ↑ | ECE ↓ | Brier ↓ | AUROC ↑ |
| 10.13 | 21.14 | 65.53 | 5.61 | 24.68 | 57.13 |
| source model: Llama2-8B | | | target model: Phi-3.8B | | |
| SciQA → TriviaQA | | | TriviaQA → SciQA | | |
| ECE ↓ | Brier ↓ | AUROC ↑ | ECE ↓ | Brier ↓ | AUROC ↑ |
| 10.56 | 20.16 | 76.04 | 11.72 | 23.04 | 68.49 |

Table 3: Cross-domain calibration within- vs. cross-backbone.

| Variants | ECE ↓ | Brier ↓ | AUROC ↑ |
| --- | --- | --- | --- |
| In-domain (w/o EM) | 6.17 | 19.25 | 75.69 |
| In-domain (w/ EM) | 4.96 | 18.39 | 78.22 |
| Cross-domain (Generalisation) | 15.91 | 22.14 | 68.96 |
| Cross-domain (Self-learning) | 17.12 | 26.55 | 64.26 |
| Cross-domain (Target EM) | 25.38 | 30.66 | 52.68 |
| Cross-domain (w/o $c_i^t$) | 13.57 | 25.71 | 65.39 |
| Cross-domain (w random rank) | 31.52 | 33.53 | 60.36 |
| Cross-domain (Ours Full) | **9.60** | **20.85** | **71.15** |

Table 4: Ablation study on TriviaQA calibration.

**Implement Details.** Our experiments consist of two stages: source-domain pretraining and target-domain adaptation. In both stages, we keep the backbone model frozen and only fine-tune a set of LoRA modules together with an independent calibration head, which greatly reduces computational overhead. We adopt a learning rate of $1 \times 10^{-5}$ and use the AdamW optimiser. The source-domain stage is trained for 3 epochs with 3 EM cycles, while the target-domain adaptation stage is trained for 3 epochs. The batch size is fixed to 16. For both backbone models, LoRA is applied to the MLP blocks, with a rank $r = 32$ and scaling factor set to 16. During source-domain training, the temperature is set to the default value of 1, while in target-domain adaptation the temperature is reduced to 0.75 and the interpolation factor $\alpha$ is set to 0.6.

### 5.2 Results and Analysis

**Results.** Tab. 1 and Tab. 2 report the experimental results using *Phi-3 3.8B* and *Llama-2 7B* as backbones. The results show that although supervised methods achieve strong calibration performance in the in-domain setting, current state-of-the-art method (Si et al., 2022; Zhao et al., 2021) suffer severe degradation when directly generalised to target domains with distribution shifts. In contrast, our method performs effective adaptation on the unlabelled target domain, leading to substantial improvements in cross-domain calibration. Remarkably, its performance approaches that of state-of-the-art supervised methods under in-domain calibration, highlighting the effectiveness of our approach. Furthermore, consistent trends are observed across both backbone models, demonstrating the robustness of our method.

**Reliability Diagrams.** In Fig. 3, we report the reliability diagrams for the calibration task on TriviaQA. The $x$-axis denotes the predicted confidence intervals, with the blue bars representing the empirical accuracy within each bin, the red line showing the sample distribution, and the dashed diagonal line indicating perfect calibration. Fig. 3(a) presents the in-domain supervised calibration result of our method on TriviaQA, where no domain shift is involved, leading to well-aligned calibration. Fig. 3(b) and (c) show the cross-domain generalisation results when training on SciQA and evaluating on TriviaQA using Apricot and our method, respectively. Due to the significant domain gap, most test samples are concentrated around 0.5 confidence, where the model is overconfident in the 0.4–0.7 range while underestimating outputs near both ends. Finally, Fig. 3(d) demonstrates our cross-domain adaptation result, where calibration performance is substantially improved compared

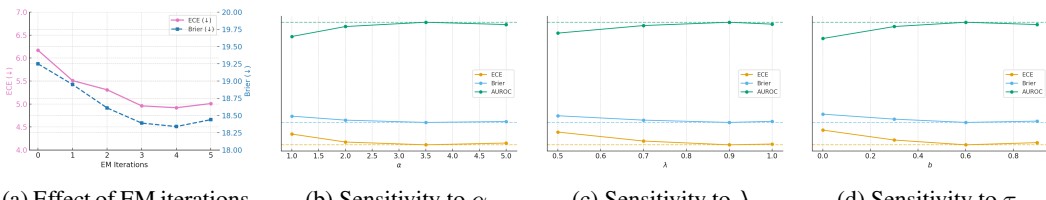

(a) Effect of EM iterations.    (b) Sensitivity to $\alpha$.    (c) Sensitivity to $\lambda$.    (d) Sensitivity to $\tau$.

Figure 4: Analysis of EM refinement and hyperparameter sensitivity. (a) Effect of EM iterations. (b)–(d) show sensitivity curves for $\alpha$, $\lambda$, and $\tau$, respectively.

to Fig. 3(b)–(c). The calibration head better assesses the reliability of the model's predictions after adaptation, which is also reflected by the lower calibration error (9.60% vs. 12.78%/15.91%).

**Cross-Backbone Calibration.** Tab. 3 compares cross-domain calibration within the same backbone (Phi-3.8B → Phi-3.8B) and across different backbones (Llama2-8B → Phi-3.8B). In both directions (SciQA → TriviaQA and TriviaQA → SciQA), RAPCal consistently improves calibration across backbones. Notably, when transferring from Llama2-8B to Phi-3.8B, performance remains competitive with the within-backbone setting, showing that calibration ability can generalise beyond architectural differences. This demonstrates that our method does not rely on strict backbone matching, making it practical for real-world scenarios where source and target models may differ.

**Ablation Study.** Tab. 4 presents the ablation results. Rows 1–2 show in-domain supervised calibration on TriviaQA. Without EM (row 1), the model relies on one-hot correctness labels, while EM refinement (row 2) provides bin-level soft labels and yields stronger calibration, highlighting the benefit of fine-grained supervision. Rows 3–5 evaluate cross-domain transfer from SciQA to TriviaQA. Direct generalisation (row 3) suffers severe degradation under distribution shift. Using calibration-head outputs for naive self-learning (row 4) further deteriorates performance due to noisy pseudo labels. Applying EM iterations in the target domain (row 5) also fails, since noise is repeatedly reinforced. Finally, rows 6–8 compare different adaptation strategies. Excluding the source calibration signals $c_i^t$ when training in the target domain (row 6) degrades performance, since pseudo labels rely solely on rank-based shaping. Replacing rank-based refinement with randomly permuted ranks (row 7) further harms performance, confirming that the gain comes from informative ranking rather than just scattering probabilities. The full model (row 8), which combines rank-based refinement with source calibration signals, achieves the best results by effectively suppressing label noise and delivering clear improvements in the cross-domain unlabelled setting.

**Hyper-parameters Analysis.** Fig. 4(a)- 4(d) analyses the impact of hyper-parameters on performance. Fig. 4(a) examines the number of EM iterations in source pre-calibration on the TriviaQA dataset, with calibration measured by ECE (lower is better). Increasing the number of iterations progressively improves calibration, and the effect stabilises after three rounds. To balance performance and efficiency, we set the number of EM iterations to 3 in all experiments. Fig. 4(b)- 4(d) evaluate sensitivity to hyper-parameters in the target domain on the SciQA → TriviaQA task. We assess three hyper-parameters: $\alpha$, $\tau$ and $\lambda$. For each evaluation, the other parameters are fixed to their optimal values. The final chosen configuration is $\alpha = 3.5$, $\tau = 0.6$, and $\lambda = 0.9$. We report results using ECE, Brier score, and AUROC. Encouraging confidence scores to spread towards the extremes improves calibration at first, but the benefit plateaus once hyper-parameters exceed a certain threshold.

## 6 CONCLUSION

This work introduces Rank-Preserving Adaptive Pseudo-Calibration (RAPCal) for black-box cross-domain QA calibration. Unlike contemporary inference confidence calibration methods that assume white-box access or labelled target data, RAPCal operates solely on model outputs and transfers calibration ability from a labelled source domain to an unlabelled target domain. By combining an EM-based fine-grained calibration strategy in the source domain with a rank-preserving refinement mechanism in the target domain, RAPCal overcomes confidence collapse under both domain and model shifts. Extensive experiments across diverse QA benchmarks demonstrate that RAPCal outperforms state-of-the-art on cross-domain calibrations, highlighting its robustness and practicality.

**Limitations.** Despite its effectiveness, RAPCal still assumes that source-trained ranks remain informative under target shift. This may not hold under extreme distribution gaps. Future work will explore adaptive shaping functions and broader evaluations in real-world QA scenarios.

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

# 7 APPENDIX

## A CODE RELEASE

Our codes are uploaded and in supplemental materials.

## B ALGORITHM FLOW

---

**Algorithm 1:** RAPCal: Rank-Preserving Adaptive Pseudo-Calibration

---

**Input:** Source dataset $\mathcal{D}_s = \{(X_i^s, \hat{Y}_i^s, l_i^s)\}$, target dataset $\mathcal{D}_t = \{X_i^t, \hat{Y}_i^t\}$, frozen backbone $f_\theta$, calibration head $h_\phi$ (2-layer MLP).

**Stage 1: Source Pre-Calibration with EM Refinement**;
**for** *iteration* $t = 0, 1, \ldots, T_s$ **do**

    **if** $t = 0$ **then**

        Train calibration head $h_\phi$ on one-hot labels $l_i^s$::;

            $c_i^s = h_\phi(f_\theta(X_i^s))$;

            $\mathcal{L}_C = \sum(c_i^s - l_i^s)^2$

    **else**

        // E-step:  estimate soft labels

        Partition $\{c_i^s\}$ into bins by value;

        For each bin $b$: $\tilde{l}_i^s = \text{Acc}(b(i))$;

        // M-step:  update calibration head

        Update $h_\phi$ with refined labels:;

            $\mathcal{L}_C = \sum(c_i^s - \tilde{l}_i^s)^2$;

    Joint loss with QA preservation:;

        $\mathcal{L}_s = \mathcal{L}_C + \gamma \mathcal{L}_{qa}(X^s, \hat{Y}^s)$;

**Stage 2: Rank-Based Pseudo-Label Construction**;
For each target prediction $c_i^t = h_\phi(f_\theta(X_i^t))$;
Compute empirical rank: $r_i^t = \frac{\text{rank}(c_i^t) + 0.5}{|\mathcal{D}_t|}$;
Apply U-shaped refinement:;

$$p_{i,\text{shaped}}^t = \begin{cases} 0.5(2r_i^t)^\alpha, & r_i^t < 0.5 \\ 1 - 0.5(2(1 - r_i^t))^\alpha, & r_i^t \geq 0.5 \end{cases};$$

Interpolate with raw confidence:;

    $\tilde{l}_i^t = \lambda c_i^t + (1 - \lambda)p_{i,\text{shaped}}^t$;

**Stage 3: Target Unsupervised Calibration**;
Initialise a new calibration head $h_{\phi'}$;
For each $X_i^t$:;

    $z_i^t = h_{\phi'}(f_\theta(X_i^t))$;

    $\hat{c}_i^t = \sigma\left(\frac{z_i^t + \tau}{T}\right)$;

Optimise with pseudo labels and QA loss:;

    $\mathcal{L}_t = \sum(\hat{c}_i^t - \tilde{l}_i^t)^2 - \gamma \sum \log p_\theta(\hat{y}_i^t | X_i^t)$;

**Output:** Adapted calibration head $h_{\phi'}$ for $\mathcal{D}_t$.

---

## C DISCUSSION ON NOT COMPARED PREVIOUS CROSS-DOMAIN CALIBRATION PAPERS.

Prior work has explored calibration in cross-domain *classification* tasks, such as TransCal (Wang et al., 2020), TransCal++ (Chen et al., 2021), and other transfer-based methods (Guo et al., 2017; Obert et al., 2021). Their main idea is to jointly exploit labelled source data and unlabelled target data by aligning confidence distributions or learning domain-invariant calibration functions, often

via domain classifiers or discrepancy minimisation. While effective for closed-set classification, these approaches are not directly applicable to LLMs in open-ended scenarios. First, they assume a shared label space across domains, whereas QA outputs are free-form text without fixed classes, making domain alignment ill-defined. Second, these methods rely on white-box access to model parameters or domain-adversarial training, which conflicts with our black-box assumption where the source model is frozen and only outputs are available. Finally, the scale and generative nature of LLMs exacerbate overconfidence differently from classification models, requiring specialised treatment. For these reasons, we acknowledge existing works but do not include them in our comparison, and instead focus on methods tailored to black-box cross-domain QA calibration.

# D    DETAILED THEORY ANALYSIS

Throughout, for domain $d \in \{s, t\}$ and input $x$, the calibration head outputs $c_d(x) \in [0, 1]$. Correctness is $C \in \{0, 1\}$ and the (target) correctness probability is $\eta_t(x) = \mathrm{Pr}_{\mathcal{D}_t}(C = 1 \mid x)$. On $\mathcal{D}_t$, for samples $\{x_i\}_{i=1}^N$ we write $c_i^t = c_t(x_i)$ and define empirical ranks

$$r_i^t \; := \; \frac{\mathrm{rank}(c_i^t) + 0.5}{N} \in (0, 1).$$

Our rank shaping uses the one-parameter map

$$h_\alpha(r) \; = \; \begin{cases} \frac{1}{2}(2r)^\alpha, & r < \frac{1}{2}, \\ 1 - \frac{1}{2}\big(2(1 - r)\big)^\alpha, & r \geq \frac{1}{2}, \end{cases} \qquad \alpha \geq 1,$$

and pseudo-labels on $\mathcal{D}_t$ are $\tilde{l}_i^t = \lambda c_i^t + (1 - \lambda) h_\alpha(r_i^t)$ with $\lambda \in [0, 1]$.

## D.1    A. RANK-SPACE TOOLKIT

**Lemma D.1** (Probability integral transform; order equivalence). *Let $Q := c_t(X)$ for $X \sim \mathcal{D}_t$ and $F_t$ be the CDF of $Q$. Then $R := F_t(Q) \sim \mathrm{Unif}(0, 1)$, and $Q$ and $R$ induce the same ordering: $Q(x_i) \leq Q(x_j)$ iff $R(x_i) \leq R(x_j)$. Moreover, any nondecreasing post-processing of $Q$ can be written as a nondecreasing function of $R$.*

**Theorem D.2** (AUC invariance under strictly monotone maps). *If $s_2 = \phi \circ s_1$ with strictly increasing $\phi$, then $\mathrm{AUC}(s_2) = \mathrm{AUC}(s_1)$.*

## D.2    B. RANK–PUSH: CONSTRUCTION AND PROPERTIES

**Proposition D.3** (Properties of $h_\alpha$). *For any $\alpha \geq 1$, $h_\alpha$ is strictly increasing, symmetric about 0.5 ($h_\alpha(r) = 1 - h_\alpha(1 - r)$), and satisfies $h_\alpha(0) = 0$, $h_\alpha(0.5) = 0.5$, $h_\alpha(1) = 1$. It is $C^1$ at 0.5 with $h_\alpha'(0.5) = \alpha$, and is $\alpha$-Lipschitz on $[0, 1]$. Moreover, $r < 0.5 \Rightarrow h_\alpha(r) \leq r$ and $r > 0.5 \Rightarrow h_\alpha(r) \geq r$ (mid-range dilution, end-range push).*

## D.3    C. SOURCE-DOMAIN EM REFINEMENT

**E/M mechanism.** On $\mathcal{D}_s$, after initial training with $l_i^s \in \{0, 1\}$ we bin by current scores $c_s(x)$, and in each bin $B_k$ compute the empirical accuracy $\widehat{\theta}_k = \frac{1}{|B_k|} \sum_{i \in B_k} \mathbb{I}\{C_i = 1\}$. We then refit the calibration head by MSE to soft targets $\tilde{l}_i^s = \widehat{\theta}_{b(i)}$.

**Lemma D.4** (Bin-wise MLE/LS estimator). *Within a fixed bin $B_k$, labels are Bernoulli with parameter $\theta_k$; the empirical accuracy $\widehat{\theta}_k$ is the MLE and also minimises $\sum_{i \in B_k}(y_i - \theta)^2$ over $\theta \in [0, 1]$.*

**Proposition D.5** (Monotone surrogate improvement under fixed partition). *Let $c_\vartheta(x)$ denote the calibration head with parameters $\vartheta$, and*

$$\mathcal{L}_\mathcal{B}(\vartheta) \; = \; \frac{1}{n} \sum_i \big(c_\vartheta(x_i) - \tilde{l}_i^s\big)^2.$$

*For a fixed partition $\mathcal{B}$, the M-step update $\vartheta^{(t+1)} = \arg\min_\vartheta \mathcal{L}_\mathcal{B}(\vartheta)$ satisfies $\mathcal{L}_\mathcal{B}(\vartheta^{(t+1)}) \leq \mathcal{L}_\mathcal{B}(\vartheta^{(t)})$, with equality iff $\vartheta^{(t)}$ is optimal.*

**With moving partitions.** When bins are recomputed after each update, the procedure becomes block-coordinate descent on a piecewise-constant surrogate defined by the current partition; empirically the objective is monotone up to partition changes and progressively replaces $\{0, 1\}$ with probability-valued supervision.

### Summary of the Detailed Theory Analysis

**A** develops the rank-space toolkit (PIT and AUC invariance). **B** formalises Rank–Push and its properties. **C** justifies the source-domain EM refinement via bin-wise MLE and a monotone surrogate decrease under fixed partitions.

## E   Reproducibility Statement

We take reproducibility seriously and have taken the following steps to ensure that our results can be verified:

- **Code and Implementation.** We provide the full implementation of our method, including training scripts, evaluation scripts, and instructions for reproducing all experiments. The code will be released in the supplementary material and made publicly available upon publication.

- **Datasets.** All datasets used in this paper (TriviaQA, SciQA, etc.) are publicly available. We include detailed preprocessing steps and data splits in the supplementary material to facilitate replication.

- **Hyperparameters.** We report all hyperparameters used in both source-domain precalibration and target-domain adaptation. Sensitivity analyses for key hyperparameters ($\alpha$, $\lambda$, $b$) are included in Section 5.

With these resources and details, we believe our results are fully reproducible by independent researchers.

## F   The Use of LLM

In preparing this submission, we made limited use of publicly available large language models (LLMs) such as ChatGPT. Their use was restricted to language refinement, including grammar correction, sentence rephrasing, and improving clarity of exposition. No LLMs were used to generate research ideas, design methodology, conduct experiments, or create results. All technical contributions, implementations, and analyses presented in this paper are solely the work of the authors.

