# OpenReview forum: "Rank-Preserving Calibration of LLMs Under Model and Distribution Shifts"
_ICLR.cc/2026/Conference — ICLR 2026 Conference Withdrawn Submission_

### Official Review · Reviewer_VWDo · 2025-10-25

**Soundness:** 3
**Presentation:** 3
**Contribution:** 2
**Rating:** 4
**Confidence:** 4

**Summary:**

This paper proposes RAPCal (Rank-Preserving Adaptive Pseudo-Calibration), a two-stage framework for calibrating LLMs under both domain and model shifts, even when target-domain labels are unavailable and the source model is black-box.
RAPCal first performs source pre-calibration using an EM-based refinement that converts binary correctness labels into soft calibration supervision. It then performs target unsupervised calibration by transforming the rank order of predicted confidences into pseudo soft labels via a U-shaped mapping function, enforcing rank consistency across shifts. Empirical results on open-ended QA datasets (TriviaQA → SciQA, etc.) show consistent improvements in ECE, Brier score, and AUROC across model backbones (Llama2–Phi3). The method is theoretically grounded via a rank-stability analysis and achieves label-free calibration that maintains task accuracy.

**Strengths:**

1. provide a framework to learn a calibrated confidence
2. good results

**Weaknesses:**

1. There are only two benchmark included, this can cause several problems.
- Authors mention that ranking preserve the uncertainty. However, is this hold for all domain shift benchmarks? We can hardly know with one domain shifted dataset results.
- like previous domain adaption methods, the adaptation should be verified on multiple OOD benchmark
2. The Multiple Choice QA performance not included. This is important when people use LLM for a classification task, which is often more related to traditional calibration. I agree the open QA is important, but Multiple Choice QA benchmarks should also be needed.
3. Does this method keep the accuracy? this should be verified on more datasets.
4. The design framework is a little bit complex, do we really need that many trainable networks? is there some better signal other than ranking? logit gap might be a good choice for domain shift, see [1]
5. I will vote yes if you could extend the experiments to a scale that is more reasonable.

[1] Sample Margin-Aware Recalibration of Temperature Scaling

**Questions:**

1. Why does the lora exist, I don't quite get it. You mentioned in a black box scenario.

---

### Official Review · Reviewer_DwKq · 2025-10-30

**Soundness:** 3
**Presentation:** 2
**Contribution:** 2
**Rating:** 4
**Confidence:** 4

**Summary:**

This paper investigates the problem of cross-domain confidence calibration for large language models (LLMs) in open-ended QA tasks. It proposes a two-stage calibration framework called Rank-preserving Adaptive Pseudo-Calibration (RAPCal). In the first stage, RAPCal uses labeled data from the source domain to pre-calibrate the LLM through an EM-based refinement process, after which the calibrated model is provided to the unlabeled target domain as a black box. In the second stage, RAPCal constructs pseudo target labels based on the relative ranking information of the initial target predictions from the source-calibrated LLM. Finally, RAPCal trains a new calibration head using the pseudo-labeled target data. Experiments involving two types of LLMs and two cross-domain QA tasks demonstrate the effectiveness and advantages of RAPCal.

**Strengths:**

- The black-box model calibration setting is practical but challenging, especially for realistic LLM applications. This paper’s investigation of this problem is valuable and should be encouraged.

- The proposed RAPCal framework is intuitively sound and general, capable of addressing both dataset shift and model shift.

- Experiments and ablation studies on two open-ended QA benchmarks demonstrate the effectiveness of RAPCal and its advantages over existing LLM calibration baselines.

**Weaknesses:**

- Literature review is incomplete and partially inaccurate. Since the paper investigates the dataset shift problem in the context of LLMs, the discussion of relevant topics such as domain adaptation should be accurate and comprehensive. However, several key related works are either misrepresented or omitted.
First, in Lines 142–144, “More recent studies in black-box domain adaptation (BBDA) (Liang et al., 2020a; Kundu et al., 2020)” is inaccurate. The two cited works are actually pioneering studies on source-free domain adaptation, which corresponds to a white-box setting where the source data is hidden from the target domain but the source model is accessible.
In contrast, works such as [1, 2] actually address the black-box domain adaptation setting that aligns with this paper’s focus—where the source model is only available for inference on target data.
Second, a previous ICML paper [3] proposed PseudoCal to tackle cross-domain calibration in both white-box and black-box domain adaptation settings. This relevant work should be discussed, especially in relation to Line 146, which currently states that “calibration under domain shift has received little attention.”

- Hyperparameter tuning issue. RAPCal involves multiple hyperparameters (Lines 470–472 and Figure 4), each of which must be carefully tuned to achieve effective calibration. However, the studied setting is highly challenging, as the target domain is entirely unlabeled and the source data is inaccessible. It remains unclear how the authors determine reasonable or near-optimal values for these hyperparameters in an unsupervised way when applying RAPCal to new cross-domain or cross-model calibration tasks.

- Practicality concern. RAPCal requires a standard pre-calibration pipeline on the source domain, which may be impractical in real-world scenarios—users typically cannot request model vendors to perform custom pre-calibration for their released LLM APIs.

- Experimental scope. The current experiments are limited to two QA benchmarks. To better demonstrate the generality and robustness of RAPCal, evaluations on additional benchmarks or diverse tasks would be beneficial.

References:

- [1] DINE: Domain Adaptation from Single and Multiple Black-box Predictors. CVPR 2022
- [2] Unsupervised Domain Adaptation of Black-Box Source Models. arXiv 2021
- [3] Pseudo-Calibration: Improving Predictive Uncertainty Estimation in Unsupervised Domain Adaptation. ICML 2024

**Questions:**

In addition to the main concerns mentioned in the weaknesses, I note the following question about the presentation:

- Figure 3 (c) and (d) are difficult to distinguish because their only difference lies in the terms “generalization” and “adaptation.” The caption should be made clearer and more detailed.

- Table 3 is unclear due to the absence of baseline results, such as those without adaptation.

---

### Official Review · Reviewer_5WWD · 2025-10-30

**Soundness:** 2
**Presentation:** 2
**Contribution:** 1
**Rating:** 2
**Confidence:** 3

**Summary:**

The paper proposes RAPCal, a rank-preserving calibration method for LLMs under distribution shifts. The method aims to improve calibration without requiring labeled target samples and is evaluated on open-domain QA transfer tasks. Results show improved calibration metrics compared to baseline approaches, though the evaluation focuses on a narrow QA setting and uses coarse sequence-level representations. Overall, the work extends traditional calibration ideas to LLM adaptation and provides preliminary evidence of effectiveness in limited scenarios.

**Strengths:**

- Proposes a simple, computationally efficient adaptation method applicable to black-box LLMs.
- Extends traditional calibration concepts to the LLM setting.
- Empirically improves calibration metrics.

**Weaknesses:**

1. The paper motivates calibration as a way to mitigate hallucination, but never measures hallucination in the experiments. In fact, it is not clear to me how a calibrated model can have less hallucinations.
2. The setting is extremely niche. It's basically a binary-classification, source-free UDA problem framed under LLM+calibration setting. I do not think this setting is impactful enough.
3. The confidence head is an MLP on the mean hidden state across the generated answer tokens (averaged over token embeddings) and outputs a single confidence score. This discards sequence structure, token-level uncertainty, and decoding dynamics; it’s unclear that such a blunt summary can generalize across prompts, lengths, or style, and hence I do not see this being practically useful.
4. Target-stage performance depends on multiple hyperparameters ($\alpha,\beta,\tau$). While the sensitivity curves are shown, how are these chosen without peeking at correctness in a label-free target? There is no unsupervised selection criterion.
5. Writing & quality issues. Unclear sentence: "resulting in adopting in-domain calibration adaptation invalid". "reliability diagrams" unclear. 'black box' uses single quotation. Notation $y_{i1},\ldots,y_{im_i}$ is confusing. Figure 3 and 4 are too small to read.

**Questions:**

- Could you provide empirical evidence as well as explanation about calibration reducing hallucinations?

---

### Official Review · Reviewer_shSu · 2025-11-02

**Soundness:** 1
**Presentation:** 3
**Contribution:** 2
**Rating:** 2
**Confidence:** 4

**Summary:**

This paper addresses confidence calibration for large language models (LLMs) under model and distribution shifts, where calibration performance often deteriorates.
The proposed method, RAPCal, trains a lightweight calibration head on labeled source-domain data via an EM-like refinement and adapts it to a target domain without labels through rank-preserving pseudo-labeling.
Experiments on QA datasets (TriviaQA, SciQA) using Llama2 and Phi-3 demonstrate improved ECE, Brier Score, and AUROC over existing baselines.

**Strengths:**

1. This paper tackles calibration under domain/model shift, an under-explored but practically important scenario.
2. The idea that relative ranking of confidence is more stable than absolute values is interesting and supported both empirically and theoretically (rank-flip bound).

**Weaknesses:**

1. The proposed EM-based calibration procedure closely mirrors the soft-label smoothing mechanism introduced by Liu et al. (EMNLP 2024), which also computes bin-level expected accuracies and iteratively refines confidence targets. The only difference lies in the explicit EM formulation and the addition of rank-based weighting. The authors should clarify whether this represents a conceptual reinterpretation of that method or a genuinely novel optimization scheme.
2. Several recent works (e.g., Liu et al., 2024; Kapoor et al., 2024) already employ a linear head on frozen LLM activations to predict confidence under soft-label or ECE-based supervision. The paper lacks the necessary citations and discussion of these related approaches.
3. The experiments compare only with older logit-based methods (Platt Scaling, Temperature Scaling, Apricot). Without including recent activation-based, head-based, or answer-based baselines (e.g., Ulmer et al., 2024), the reported improvement margins are not convincing.
4. In Table 1/2, the paper cites three methods, Seq. Likelihood (Kamath et al., 2020), Apricot (Zhao et al., 2021) and ActCab (Si et al., 2022), as baseline approaches.
However, after searching through ACL 2020, NeurIPS 2021/2022 proceedings, arXiv, and Google Scholar, I was unable to locate any published papers matching these titles, authors, or venues:
- Kamath et al. (2020) “Selectivity in Question Answering: No Free Lunch in QA” (claimed ACL 2020)
- Zhao et al. (2021) “Calibrating Language Models with Multiple Choice Questions” (claimed NeurIPS 2021)
- Si et al. (2022) “Actively Calibrated Selective Classification” (claimed NeurIPS 2022)

Could the authors please clarify the provenance of these citations?
If these works are unpublished, internally circulated, or fabricated placeholders (e.g., generated by an LLM or citation autocomplete), that should be clearly stated and corrected.
Currently, these references undermine the credibility of the reported baselines.

5. The study evaluates only two datasets (TriviaQA and SciQA). Given the paper’s central claim of domain/model-shift robustness, this limited scope substantially weakens the generality of the conclusions.


Reference:
1. Liu et al., 2024, Enhancing Language Model Factuality via Activation-Based Confidence Calibration and Guided Decoding.
2. Kapoor et al., 2024, Large Language Models Must Be Taught to Know What They Don't Know.
3. Ulmer et al., 2024, Calibrating Large Language Models Using Their Generations Only.

**Questions:**

1. How is the EM refinement in Algorithm 1 different from the “expected confidence label” approach used in ActCab (Liu et al., 2024)?
If it is mathematically equivalent, please clarify and cite accordingly.

**Details Of Ethics Concerns:**

The paper cites several works that appear to be non-existent or unverifiable in the claimed venues. Specifically,
- Kamath et al. (2020) “Selectivity in Question Answering: No Free Lunch in QA” (claimed ACL 2020),
- Zhao et al. (2021) “Calibrating Language Models with Multiple Choice Questions” (claimed NeurIPS 2021), and
- Si et al. (2022) “Actively Calibrated Selective Classification” (claimed NeurIPS 2022)

cannot be found in the official ACL or NeurIPS proceedings, arXiv, or Google Scholar.

These fabricated or unverifiable references call into question the validity of the reported baselines and raise potential research integrity concerns.
An ethics review may be necessary to verify the authenticity of these citations and ensure that the experiments were conducted on real, published baselines.

---

### Note · Authors · 2025-11-12

I have read and agree with the venue's withdrawal policy on behalf of myself and my co-authors.